# "Disorder" versus "Abuse"? Exploratory Data on Stigmatizing Terminology among Medical Students at a Swiss University

Manon Baehler [1,*], Emilien Jeannot [2,3], Deborah Lidsky [4], Gilles Merminod [5], Cheryl Dickson [2] and Olivier Simon [2]

1. Faculty of Biology and Medicine, University of Lausanne, Rue du Bugnon 21, CH-1011 Lausanne, Switzerland
2. Centre du jeu Excessif, Addiction Medicine, Lausanne University Hospital, University of Lausanne, Avenue de Morges 10, CH-1004 Lausanne, Switzerland; emilien.jeannot@chuv.ch (E.J.)
3. Faculty of Medicine, Institute of Global Health, Chemin de Mines 9, CH-1202 Geneva, Switzerland
4. Dependencies Unit, Division for Primary Care, Geneva University Hospitals, Boulevard de la Cluse 75, CH-1205 Geneva, Switzerland
5. Department of French, Faculty of Arts, University of Lausanne, Quartier UNIL-Chamberonne, CH-1015 Lausanne, Switzerland
* Correspondence: manon.baehler@unil.ch; Tel.: + 41-79-675-40-53

**Abstract:** The study of wording and its impact on medical practice is key for the training of future physicians. Negative, imprecise, and disrespectful terms are still widespread in the medical field and contribute to the stigmatization of people in treatment, which ultimately limits their access to care. In this study, we explore the feasibility and acceptability of a method to investigate medical students' perceptions of wording and stigma. This method involves a questionnaire that medical students complete after having read a clinical vignette. One of the two versions of the vignette is made available, which only varies in the way the patient is referred to ("substance abuser" vs. "having a substance use disorder"). Medical students from the University of Lausanne between their first and sixth year were contacted via the university's mailing lists. They were randomly exposed to one of the two versions of the vignette and responded to the questionnaire online. This exploratory study shows that it is feasible and acceptable to assess the influence and perceptions of stigmatizing terminology among students through a vignette-based questionnaire comparing two wording options. In line with the initial study, we find trends in favor of the non-stigmatizing terminology; however, beliefs are widely held about the need for judicial "punishment" to address consumption behavior. No statistically significant differences are found between the two groups. The study of wording and its impact on access to care is a crucial issue which seems necessary to integrate into pre-graduate training. It permits the deconstruction of prejudices related to medical knowledge and offers perspectives for intervention and research to improve the right to health, which includes the fundamental right to access to care.

**Keywords:** terminology; stigmatizing language; person-first; person-centered

## 1. Introduction

Stigma can be described as a dynamic process of devaluation that significantly discredits an individual in the eyes of others, such as when certain attributes are seized upon within particular cultures or settings and defined as discreditable or unworthy. According to Link and Phelan's conceptualization, stigma exists when elements of labelling, stereotyping, separating, status loss, and discrimination co-occur in a power situation that allows these processes to unfold [1]. This conceptualization suggests that stigma is likely to play a key role in determining social opportunities, and may be linked to social withdrawal, inequitable healthcare access, poorer physical health, employment, housing, and general lifestyle. When stigma is acted upon, the result is discrimination, as shown in the World

Drug Report of 2018 [2], causing feelings of rejection, shame, and reinforcing social isolation, preventing in turn ill persons from seeking access to medical care [3].

In the healthcare field, it can present itself in many forms, as outlined in the Box 1 below.

**Box 1.** Definition of stigma.

> Stigma has been studied many times over the years. Following its conceptualisation [1,4] it has been redefined by Lancaster et al., [3] as follows:
> *"Stigma is labelling and stereotyping of difference, at both an individual and structural societal level, that leads to status loss (including exclusion, rejection and discrimination). It leads to discrimination which is the lived effects of stigma—the negative material and social outcomes that arise from experiences of stigma. Both of those concepts rely on societal structures and systems that facilitate and create the conditions for their operation. "*
> Different forms of stigma are distinguished: Perceived-, Enacted-, and Self- Stigma:
> *Perceived stigma refers to beliefs that members of a stigmatized group have about the prevalence of stigmatizing attitudes and actions in society.*
> *Enacted stigma refers to directly experienced discrimination based on membership in a stigmatized group.*
> *Self-stigma refers to negative thoughts and feelings (e.g., shame, negative self-evaluative thoughts, fear) that emerge from identification with a stigmatized group and their resulting behavioral impact avoidance of treatment, failure to seek employment, and avoidance of intimate contact with others* [5].

For example, stigma can be felt by the people in treatment, coming from the healthcare professionals through their attitude or comments, can be the source of bias that alters the way professionals act towards certain groups of people [6], and can be found in notes and medical records, scientific articles, and other documents [7–9]. Many groups of people in treatment are stigmatized. Personal and social distinctions such as race/ethnicity, gender, age, phenotype, and behavior can lead to stigma and discrimination. As substance-use disorder (SUD) is often viewed as a moral or criminal issue, rather than as a health issue, people with SUD are particularly at risk of structural discrimination. People who have a SUD may come from different backgrounds; for example, they may use mental health inpatient or outpatient services, belong to different ethnic groups, or be people who consume non-medical substances and alcohol. Most of the time the populations concerned are already marginalized and access to health can already be an issue for them [10]. In this article, we choose to focus on people with SUD. This population is constantly exposed to stigmatizing attitudes coming from their family, their friends, the general population, and sometimes, healthcare establishments/staff [11–13].

Previous research has highlighted that many of the people concerned use strategies to delay or avoid care because they had been treated unfairly or discriminated against in the past. Such discrimination can present itself by the person being treated differently from other people and/or feeling like the staff "looked down on them", not having their complaint taken seriously, having difficulty getting pain medication because of their history of dependence, or even being labelled, for example, as "junkies; only here to score some opiates" [12–14]. Some people develop strategies such as delaying the search for care, not mentioning their drug use, downplaying their need for pain medication, and sometimes seeking alternative services to try to avoid these situations [15]. Consequently, all of these experiences lead to a treatment gap between people with SUD and other patients, and affect the quality of care they receive from professionals [16–20].

Stigma and discrimination can present themselves in multiple forms. Among these, the terminology physicians use on an everyday basis—even if it might pass by unnoticed at first glance—has an impact on patients. If people with a substance-use disorder are exposed every day to all kinds of demeaning, judgmental, and stigmatizing words when interacting with others, it can also occur while they are cared for in health establishments [21,22]. Words such as "substance abuser", "addict", a "clean" or "dirty" blood sample, and "substitution" treatment are often used amongst healthcare professionals. [23]. The use of these words is problematic because they are associated with feelings, impressions or negative connotations that affect the way we perceive the person [24]. Words are not just

labels that we put on a pre-existing reality; they are tools that organize our perception of the world and influence our decisions and actions [25,26]. For a few years now, many journals, institutions and collectives (associations/federations) have promoted the use of a person-first/-centered language (PFL) to try and remedy this problem [27–29]: "*PFL is a way of referring to individuals with medical conditions or disability that emphasize the person over their condition or disability—is important in reducing stigma surrounding individuals*" [9]. In our case, PFL is considered for people who exhibit substance-seeking behavior.

PFL promotes description (e.g., a person with an alcohol-related problem) over labelling (e.g., an alcoholic) because the latter reduces the persons to their condition. The main objective of PFL is to avoid defining people with a single aspect of their condition to humanize and decrease the stigma. It is even more important when we refer to people with SUD because these individuals are often perceived as being responsible for causing their condition and having control over their consumption. Some of the terms typically used to describe their situation reinforce this idea, such as "the patient confesses to drinking" [16,27]. It is important to promote a person-first form of language along with other interventions to reduce stigma. Such an approach has proven to favor better care, better inter-professional communication, and alongside this, a more accurate knowledge and understanding of the people affected by the condition, the professionals providing their care, and the public [30,31].

One way of achieving this is to promote the approach to younger generations and especially to future healthcare professionals who are currently learning the skills of engaging and building strong therapeutic relationships with their patients. Particularly, it has been shown in the past that anti-stigmatizing programs have had an impact on the physician-to-be's practice [32], highlighting the importance of addressing stigma within healthcare services.

The main purpose of this work is therefore to explore the perceptions of future doctors around a clinical case involving substance-use disorder and touching on issues relating to care and justice.

## 2. Method

### 2.1. Participants

The participants of our study are medical students between their 1st and 6th year of study, at the University of Lausanne, during the academic year 2021–22. The average age of all students is 23.12 years old, 67% of the sample are female, and 33% are male. Regarding their learning context, a new frame of reference was introduced for the federal exam of medicine in 2021: PROFILES (*Principal Relevant Objectives and Framework for Integrative Learning and Education in Switzerland*) [1,5,33]. It includes clinical, public health, and ethical situations and promotes a higher level of medical education that tends to be closer to the reality of medical practice. It also stresses the characteristics of a fully qualified physician, which include autonomy, communication skills, professionalism, and capacity for interprofessional collaboration, as well as the ability to adopt a reflective approach toward the progress of medical science.

It is presented through three main chapters that are all interconnected, different roles of a doctor inspired from the CanMEDS roles already used worldwide (*Medical Expert* (*EXP*), *Communicator* (*COM*), *Collaborator* (*COL*), *Leader/Manager* (*LEA*), *Health Advocate* (*ADV*), *Scholar* (*SCH*), *Professional* (*PRO*), entrustable professional activities (EPAs) and situations as starting points (SSPs)) [33,34].

Within this context, we find it relevant to evaluate the current students' interest in the subject and their perceptions to raise awareness of the implications of terminology in their future practice.

### 2.2. Questionnaire

For our study we use the questionnaire developed by John F. Kelly and Cassandra M. Westerhoff [18].

This questionnaire compares the perceptions and attitude of mental health professionals toward an individual in treatment depending on the terminology used to describe his life situation. In the initial study, the questionnaire included questions previously used in two national surveys [31] alongside some additional questions that were constructed by Kelly and Westerhoff [18]. The questionnaire presented the same person but described in two different vignettes as; "a substance abuser" vs. "having a substance use disorder". (In our questionnaire, terms are translated in French as ''Toxicomane'' vs. ''ayant un trouble d'usage de substances''). They were then asked 32 questions (using a 6-point Likert-scale) that evaluate whether the person was perceived as the cause of his problem, if he was a social threat, able to regulate his substance use, and if he should receive therapeutic vs. punitive actions [18]. The French version used for this study can be seen in Appendix A. The two vignettes that have been adapted from Kelly and Westerhoff's original questionnaire [18] are also available in Appendix B.

### 2.3. Procedure

Initially, we contacted one of the authors to ask if they agreed that we could translate and use their questionnaire. With their permission, we proceeded to translate a first version of the vignettes and questions. It was then sent by email to a group of 10 students who volunteered, as a pre-test, to review our translation and see if every item was understandable when reading.

When they gave us their feedback, a few changes were made so that the survey could be validated. The principal reformulation was for the sentence of the vignette explaining that Mr. Williams had undertaken 2 urine tests and a breathalyser, and tested positive for alcohol and drug consumption. The phrasing in French was too long and confused some of the participants. We decided to structure the sentence with comas so it would be more comprehensible. We then contacted the medical students' association to obtain the mailing addresses, enabling us to pass the questions to all of the students from the faculty of medicine at the University. We loaded the questionnaire with the two vignettes and the questions onto the Limesurvey platform, which we were given access to through a university login. In addition to the original clinical vignette and 32 items, the survey contained a question asking which class the students were in (1st–6th year). The distribution between the two clinical descriptions ("*a substance abuser*" vs. "*having a substance use disorder*") was randomly assigned by the relevance equation we had configured prior to their registration. When a participant clicked on the link to answer the questionnaire, the Limesurvey platform randomly assigned one of the two versions to him or her. In October 2021, we sent the survey a first time using the mailing lists we were given earlier, and a few weeks later, sent it a second time to increase the response rate. After approximately two months, the access to the questionnaire was closed so that we could analyze the data. To encourage students to take part in our study, we offered the chance to win one of eleven cinema tickets. If interested, they could enter their personal mail address at the end of the questionnaire to take part in the draw. We did not have access to participants' socio-demographic information such as their age, sex or ethnicity, due to the fact that that the survey was distributed through a mailing list and not directly to individual students. Data were extracted in an excel sheet considering each class individually (1st–6th).

### 2.4. Sample Size

We included in our data all of the participants that answered the questionnaire between October 2021 and December 2021.

### 2.5. Statistical Analysis

We first regrouped the answers for the 6-point Likert-scale into two categories: "agree" and "disagree". We initially added the possibility to respond to the questions with an "indifferent" option in addition to the 6 points Likert-scale. However, as there were very few participants using the "indifferent" option, these responses were ultimately removed

from the analysis so that this would not influence the weight between the two previous categories. Each question was analyzed to see if there were differences between the answers of the two groups, who had been either exposed to vignette A or B. A description of the population selected and a comparison of the responses between vignettes A and B were made using the Khi square test. A *p* value of at least 0.05 was considered to be statistically significant. Statistical analyses were carried out using STATA 14.

### 2.6. Ethics & Editorial Policies

We contacted the Cantonal Commission for Ethics and Research (CER-VD) to ask whether our project should be submitted to them before passing our questionnaire. Since our project did not fall within the scope of the Law Relating to Human Research (LRH), they confirmed to us that a submission was not necessary.

## 3. Results

As shown in Table 1, we received 402 responses to our study, representing a 20.09% global response rate (194 participants for vignette A and 208 for vignette B).

**Table 1.** Number of responses and participation rates by vignette and study year.

|  | VIGNETTE A * | | VIGNETTE B * | | |
|---|---|---|---|---|---|
|  | **n** | **%** | **n** | **%** | |
| 1st year (Bmed1) | 55 | 28.35 | 57 | 27.40 | 112 |
| 2nd year (Bmed2) | 26 | 13.40 | 36 | 17.31 | 62 |
| 3rd year (Bmed3) | 23 | 11.86 | 33 | 15.87 | 56 |
| 4th year (Mmed1) | 27 | 13.92 | 27 | 12.98 | 54 |
| 5th year (Mmed2) | 39 | 20.10 | 31 | 14.90 | 70 |
| 6th year (Mmed3) | 24 | 12.37 | 24 | 11.54 | 48 |
|  | 194 | | 208 | | 402 |

* Vignette A contains the "substance abuser" and vignette B the "substance use disorder" term.

The email was sent to all medical students at the University of Lausanne, comprising *809 1st year, 273 2nd year, 240 3rd year, 243 4th year, 330 5th year and 106 6th year students (2001 in total)*. The response rates for each year group (1st–6th) are 13.84%, 22.71%, 23.33%, 22.22%, 21.21% and 45.28%, respectively. There is an important difference in the response rate for the different school years, which leads to self-selected groups of participants which are closer in size (ranging from 55 to 100 participants). The proportion of participants in each year of study and for both vignettes is presented in Table 1. A little more than half of the participants are currently in their Bachelor's years (230) and the rest (178) are in their Master's years.

A significant number of medical students had negative attitudes toward people with SUD in both groups. They consider M. Williams to be "responsible for causing his problem" (31.91%; 25.89%) and believe that "he will do something violent to others" (21.79%; 19.35%). 147 students even thought that Mr. Williams should be punished harder by the judge for his substance consumption (question 12).

We found no statistically significant ($p < 0.05$) differences in this sample as presented in Table 2. Nevertheless, we can make some general observations. The responses that are closer to a significant *p*-value are from questions 17 ("*I would be willing to have Mr. Williams as a neighbor*"), 23 ("*Mr. Williams should be prescribed medication*"), and 29 ("*He should undergo urine/breathalyzer testing/transdermal monitoring*") with a $p = 0.09$, and 25 ("*Mr. Williams' problem is caused by a chemical imbalance in the brain*") with a $p = 0.05$. In all of the questions asked to the two groups, only the first one seems to reveal different opinions between groups A and B ("*In order to help Mr. Williams stay on track, the judge should initiate disciplinary action*"), with 54.21% agreeing in group A versus 49.01% for group B. These

trends suggest the direct impact of language on preferred social distance, particularly in relation to factors such as suitable neighborhood, employment of screening, and therapeutic approach (follow-up, medication).

**Table 2.** Distribution of responses for Vignettes A et B, and comparison.

| QUESTIONS | | VIGNETTE A | | VIGNETTE B | | |
|---|---|---|---|---|---|---|
| | | N | % | N | % | *p* |
| 1 | AGREE | 103 | 54.21 | 99 | 49.01 | 0.72 |
| | DISAGREE | 87 | 45..79 | 103 | 50.99 | |
| 2 | AGREE | 101 | 56.11 | 110 | 57.89 | 0.65 |
| | DISAGREE | 79 | 43.89 | 80 | 42.11 | |
| 3 | AGREE | 82 | 43.85 | 94 | 47.24 | 0.6 |
| | DISAGREE | 105 | 56.15 | 105 | 52.76 | |
| 4 | AGREE | 19 | 10.22 | 16 | 7.92 | 0.8 |
| | DISAGREE | 167 | 89.78 | 186 | 92.08 | |
| 5 | AGREE | 10 | 5.85 | 18 | 9.94 | 0.43 |
| | DISAGREE | 161 | 94.15 | 163 | 90.06 | |
| 6 | AGREE | 58 | 30.53 | 51 | 25.89 | 0.5 |
| | DISAGREE | 132 | 69.47 | 146 | 74.11 | |
| 7 | AGREE | 60 | 31.91 | 51 | 25.89 | 0.5 |
| | DISAGREE | 128 | 68.09 | 146 | 74.11 | |
| 8 | AGREE | 2 | 1.05 | 2 | 1.04 | 0.82 |
| | DISAGREE | 188 | 98.95 | 191 | 98.96 | |
| 9 | AGREE | 7 | 3.63 | 3 | 1.49 | 0.74 |
| | DISAGREE | 186 | 96.37 | 199 | 98.51 | |
| 10 | AGREE | 65 | 39.63 | 83 | 48.82 | 0.43 |
| | DISAGREE | 99 | 60.37 | 87 | 51.18 | |
| 11 | AGREE | 39 | 21.79 | 36 | 19.35 | 0.8 |
| | DISAGREE | 140 | 78.21 | 150 | 80.65 | |
| 12 | AGREE | 76 | 40 | 71 | 35.32 | 0.5 |
| | DISAGREE | 114 | 60 | 130 | 64.68 | |
| 13 | AGREE | 90 | 52.63 | 82 | 45.56 | 0.4 |
| | DISAGREE | 81 | 47.37 | 98 | 54.44 | |
| 14 | AGREE | 110 | 63.95 | 113 | 58.55 | 0.5 |
| | DISAGREE | 62 | 36.05 | 80 | 41.45 | |
| 15 | AGREE | 69 | 38.55 | 55 | 28.80 | 0.46 |
| | DISAGREE | 110 | 61.45 | 136 | 71.20 | |
| 16 | AGREE | 98 | 57.31 | 111 | 60.66 | 0.8 |
| | DISAGREE | 73 | 42.69 | 72 | 39.34 | |
| 17 | AGREE | 134 | 75.28 | 126 | 67.02 | 0.09 |
| | DISAGREE | 44 | 24.72 | 62 | 32.98 | |
| 18 | AGREE | 119 | 66.85 | 114 | 59.07 | 0.3 |
| | DISAGREE | 59 | 33.15 | 79 | 40.93 | |
| 19 | AGREE | 182 | 95.29 | 178 | 87.68 | 0.4 |
| | DISAGREE | 9 | 4.71 | 25 | 12.32 | |
| 20 | AGREE | 180 | 96.26 | 192 | 97.96 | 0.6 |
| | DISAGREE | 7 | 3.74 | 4 | 2.04 | |

**Table 2.** *Cont.*

| QUESTIONS | | VIGNETTE A | | VIGNETTE B | | |
|---|---|---|---|---|---|---|
| | | N | % | N | % | *p* |
| 21 | AGREE | 63 | 36.21 | 59 | 32.42 | 0.2 |
| | DISAGREE | 111 | 63.79 | 123 | 67.58 | |
| 22 | AGREE | 185 | 96.35 | 199 | 98.51 | 0.25 |
| | DISAGREE | 7 | 3.65 | 3 | 1.49 | |
| 23 | AGREE | 169 | 91.35 | 170 | 86.73 | 0.09 |
| | DISAGREE | 16 | 8.65 | 26 | 13.27 | |
| 24 | AGREE | 194 | 100 | 208 | 100 | 1 |
| | DISAGREE | 0 | 0 | 0 | 0 | |
| 25 | AGREE | 123 | 71.93 | 143 | 81.25 | 0.05 |
| | DISAGREE | 48 | 28.07 | 33 | 18.75 | |
| 26 | AGREE | 188 | 98.43 | 207 | 100 | 0.4 |
| | DISAGREE | 3 | 1.57 | 0 | 0 | |
| 27 | AGREE | 84 | 46.41 | 86 | 43.22 | 0.8 |
| | DISAGREE | 97 | 53.59 | 113 | 56.78 | |
| 28 | AGREE | 11 | 5.76 | 14 | 6.86 | 0.9 |
| | DISAGREE | 180 | 94.24 | 190 | 93.14 | |
| 29 | AGREE | 115 | 63.54 | 97 | 53.59 | 0.09 |
| | DISAGREE | 66 | 36.46 | 84 | 46.41 | |
| 30 | AGREE | 107 | 60.45 | 125 | 63.13 | 0.66 |
| | DISAGREE | 70 | 39.55 | 73 | 36.87 | |
| 31 | AGREE | 134 | 85.35 | 144 | 86.75 | 0.9 |
| | DISAGREE | 23 | 14.65 | 22 | 13.25 | |
| 32 | AGREE | 128 | 73.99 | 145 | 75.92 | 0.72 |
| | DISAGREE | 45 | 26.01 | 46 | 24.08 | |

On the other hand, some questions reveal opinions that were almost unanimous for every participant. For example, 98.95% (A) and 98.96% (B) disagree that "*Mr. Williams' problem is God's will*". In the same way, 96.37% (A) and 98.51% (B) disagree that "*He should be given some kind of jail sentence as a "wake up" call*". When participants are asked if "*Mr. Williams should be referred to a therapist/psychologist/social worker*", 100% of both groups agreed. They also agree that "*He should be referred to a self-help group (e.g., AA)*" at a rate of 98.43% for group A and 100% for B.

## 4. Discussion

In line with our hypothesis, this exploratory study shows that it is feasible and acceptable to assess the influence and perceptions of stigmatizing terminology among students through a vignette-based questionnaire comparing two wording options. The access to students granted by the faculty also demonstrates the interest and encouragement of teaching staff to carry out studies within the context of the diploma course. Such undertakings enable an evaluation of the different dimensions of the university curriculum and their impact upon students.

The theme of stigma manifested by wording is a subject of interest to the participants and deserves to be further investigated since a non-stigmatizing attitude is one of the explicit objectives of the pre-graduate training [34].

The results indicate that a significant number of medical students have negative attitudes toward people with SUD. Regardless of what language is used, some participants consider the person with SUD to be responsible for their own health condition. For example, when they are asked if Mr. Williams' problem was caused by his bad choices (question 3), 44% (vignette A) versus 47% (vignette B) of participants agree. Moreover, the individual's

behavior is perceived by some as a treat, and they believe that he should be punished for his substance consumption. In this instance, 54% (A) versus 49% (B) of participants agreeing that a disciplinary action should be initiated by a judge (question 1). Such results confirm that a significant number of medical students have stigmatizing attitudes, which could lead to a moralistic approach towards people suffering from SUD, as mentioned in the majority of studies.

The importance of neutral, precise, and respectful terminology in medical communication is now a subject of consensus [21]. It is now a question of finding alternatives and ways not to reproduce practices that maintain or amplify stigmatizing attitudes in the future, particularly since they are the premise of discriminatory behavior which, de jure or de facto, carries the ultimate risk of exclusion from access to care.

With regard to our aim, we are able to explore some trends in the perceptions of future healthcare professionals. However, it is important to bear in mind that, due to the limited sample size, some results are weak or not significant and therefore only open to speculation. Kelly and Westerhoff's original study found small differences in some of the answers to the questions. Those concerning the degree of punitive action and the person's share of responsibility for the cause of his or her problem seem to further divide the participants exposed to the two terminological options (Substance abuser vs. Substance-use disorder). However, they d0 not identify differences of opinion on issues relating to the social threat posed by the person with SUD or the modalities of treatment [18].

In line with the original study, we do not observe any statistically significant differences even though the trends are similar. The only issue where the tendencies are at odds in our study is the possibility of punitive action by a judge. The question with the *p* value that is closest to being significant (*p* = 0.056) relates to the attribution of the cause of Mr. Williams' problem and shows a difference of almost 10% between the participants over the 2 vignettes: Mr. Williams' problem is caused by a biochemical imbalance in the brain (question 25), with 72% agreeing for vignette A vs. 81% for vignette B. Regarding the treatment that should be provided to Mr. Williams, we also see that the 2 groups tend to have the same opinions, as shown by the questions that suggest that he should be referred to a therapist/psychologist/social worker (question 24) or that he join the self-help group, Alcoholics Anonymous (question 26). For these questions, there was almost unanimous agreement between the participants of both groups. For questions that suggest the patient may be a social threat, the trends were not pronounced and did not necessarily suggest a bias caused by the two terminological options. Although they are not statistically reliable in our sample, we can hypothesize that these trends suggest the direct impact of language on perception, judgment, and therapeutic approach. However, a larger-scale study is needed to further explore this issue and confirm or refute our hypothesis.

## 5. Strengths and Limitations

One of the main limitations of our exploratory study is that, due to the setting of our study, we could not obtain the sociodemographic data of our sample, which does not allow us to compare the groups exposed to the two terminological options and the trends among them according to age or sex. Our results also come from a sample whose size remains modest and with a response rate of 20% of the entire population of pre-graduate students. Furthermore, there was a varied uptake rate across the academic years, ranging from 13.84% in the 1st year to 45.26% in the 6th year. It is difficult to know the reasons why certain year groups were particularly motivated to take part, and we are subsequently unaware how this will have impacted participants' responses within the study. In terms of the study design, we should also note that, due to the varied uptake rate, some year groups (e.g., year 6) were better represented than others (e.g., year 1). Another limitation concerns the questionnaire from the original study, for which we were unable to obtain the psychometric documentation/statistical key and which only benefited from a back-translation supplemented by a pre-test for its translation into French. The strengths of

our study include the interest of the participants and the topicality of this research theme, which is still very little studied.

## 6. Conclusions

The study of wording and its influence on access to care is a critical issue which appears necessary to integrate into the evaluation of pre-graduate training. In this exploratory study, the influence of terminology is measured by the impact of an important but unique term (designation of the patient); however, there are many other negative, imprecise, and disrespectful terms that can contribute to stigma among professionals [21,29]. Our recommendations for future studies would be to investigate this theme with a validated questionnaire that would test the influence and perception of a set of terminological choices in order to show statistically and clinically significant results. Such work could serve as a reference point for future monitoring efforts. In most of the studies that measure stigma among medical students, a wide range of questionnaires are used, and most of these are not validated. It would also be interesting to be able to pass such an instrument to a cohort of students in order to test their perceptions at several stages of their pre-graduate training and assess the impact of teaching. More generally, studying the influence of professional wording and the deconstruction of prejudices related to medical knowledge offers many opportunities to facilitate the right to health, which includes the fundamental right to access to care.

**Author Contributions:** Conceptualization: M.B., E.J. and O.S.; Writing, original draft, M.B., E.J., D.L., G.M., C.D. and O.S.; Formal analysis: M.B. and E.J.; Reviewing and editing: M.B., E.J., D.L., G.M., C.D. and O.S. All authors have read and agreed to the published version of the manuscript.

**Funding:** This research received no external funding.

**Institutional Review Board Statement:** As the health-related data were collected anonymously, the data does not fall under the Swiss Federal Law on Human Research.

**Informed Consent Statement:** The contents of the study and the conditions of participation were explained to participants when they opened the Limesurvey link and before answering the questions. Their consent was given by passing to the next step to read the vignettes.

**Data Availability Statement:** Data will be made available by the authors upon reasonable request.

**Conflicts of Interest:** The authors declare no conflict of interest.

## Appendix A

French version of the questionnaire used in this study.

1.  Afin de pouvoir aider M. Williams à rester sur la bonne voie, le juge devrait initier une action disciplinaire.
2.  Je crois que M. Williams va faire preuve d'une forme de violence envers lui-même.
3.  Son problème est causé par des mauvais choix qu'il a fait.
4.  M. Williams devrait se voir assigné 200 h de travail d'intérêt général.
5.  M. Williams devrait être référé à un professionnel de l'accompagnement spirituel.
6.  Son problème est causé pas un mode de vie irresponsable.
7.  M. Williams est responsable de la cause de son problème.
8.  Le problème de M. Williams est lié à la volonté divine.
9.  Il devrait se voir attribué une peine de prison comme un rappel à l'ordre.
10. M. Williams devrait être référé à un naturopathe ou un accompagnant spirituel.
11. Je crois qu'il va commettre des actes de violences envers des tiers.
12. Le juge devrait accroitre la sévérité des conséquences en cas de toute reconsommation d'alcool ou de drogue.
13. Son problème est causé par la manière dont il a été élevé.
14. M. Williams aurait pu éviter de consommer de l'alcool et des drogues.
15. Je serais prêt à avoir M. Williams comme employé.

16. Je serais prêt à avoir M. Williams comme ami proche.
17. Je serais prêt à avoir M. Williams comme voisin.
18. Je serais prêt à avoir M. Williams comme collègue de travail.
19. Afin de pouvoir aider M. Williams à rester sur la bonne voie, le juge devrait initier un traitement plus intensif.
20. M. Williams devrait parler à sa famille et ses amis au sujet de sa situation.
21. Son problème est génétique ou héréditaire
22. Il est probable que les problèmes de M. Williams soient très sévères.
23. M. Williams devrait se voir prescrire une médication.
24. M. Williams devrait être référé à un thérapeute/psychologue/assistant social.
25. Le problème de M. Williams est causé par un déséquilibre biochimique dans le cerveau.
26. Il devrait être référé à un groupe d'entraide. (p.ex AA)
27. M. Williams devrait être référé à un hôpital psychiatrique.
28. M. Williams est capable de surmonter son problème par lui-même.
29. Il devrait subir des examens d'urine/des éthylotest/une surveillance transdermique.
30. Je crois que M. Williams est capable de discernement quant à son traitement.
31. Le problème de M. Williams est causé par un contexte de stress.
32. M. Williams devrait être référé à un médecin de premier recours.

**Appendix B**

**French version of the two vignettes adapted from Kelly and Westerhoff's questionnaire [18]**

**« Vignette A—Substance abuser = Toxicomane »**

M. Williams est un toxicomane et suit un programme de traitement par l'intermédiaire du tribunal. Dans le cadre du programme, il est requis de M. Williams qu'il reste abstinent d'alcool et d'autres drogues. Il a été compliant envers les conditions du programme, jusqu'à il y a un mois, lorsqu'il a été observé deux tests positifs de toxicologie urinaire qui ont révélé une consommation de drogue et un résultat d'éthylotest qui a révélé une consommation d'alcool. Au cours du dernier mois, il y a eu un autre test de toxicologie urinaire qui a révélé l'usage de drogue. M. Williams a été un toxicomane depuis quelques années. Il attend actuellement son rendez-vous avec le juge pour déterminer son statut.

**« Vignette B—Substance Use Disorder = Trouble lié à l'usage de substances »**

M. Williams présente un trouble lié à l'usage de substances et suit un programme de traitement par l'intermédiaire du tribunal. Dans le cadre du programme, il est requis de M. Williams qu'il reste abstinent d'alcool et d'autres drogues. Il a été compliant envers les conditions du programme, jusqu'à il y a un mois, lorsqu'il a été observé deux tests positifs de toxicologie urinaire qui ont révélé une consommation de drogue et un résultat d'éthylotest qui a révélé une consommation d'alcool. Au cours du dernier mois, il y a eu un autre test de toxicologie urinaire qui a révélé l'usage de drogue. M. Williams présente un trouble lié à l'usage de substances depuis quelques années. Il attend actuellement son rendez-vous avec le juge pour déterminer son statut.

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
