# Peer review of "“Disorder” versus “Abuse”? Exploratory Data on Stigmatizing Terminology among Medical Students at a Swiss University"

_2673-5318, doi:10.3390/psychiatryint4020012_

Round 1

Reviewer 1 Report

The authors manuscript on “Disorder” versus “Abuse”? Exploratory data on stigmatizing terminology among medical students at a Swiss University “  is a well written  with clearly defined  the topic and the results obtained in this  paper. However, it is the reviewer's responsibility to draw the authors' attention to aspects requiring revision / improvement:

    First page: missing information about type of manuscript ( review or article) above the title in left part of the page.

    It seems missing  reference(s) numer(s) to support quoted definition of stigma in ,,Box 1: Definition of stigma “.

    All abbreviation used first time in the main text must be explained. In example abbreviation ,,SUD” in line 73,74, 75 is explained in line 80 (,,substance use disorder (SUD) “)but not in line 73.

    line 153-154, (quote) ,,For our study we used the questionnaire developed by John F. Kelly and Cassandra 153 M. Westerhoff “ must be supported by reference.

    Line 172-173, please be more specify about details of quoted ,, Initially, we contacted one of the authors to ask if they agreed that we could translate and use their questionnaire “. It is missing data : Who are the authors of questionnaire? What is name of questionnaire?

    It is recommended to provide and attach questionnaire used in this study to manuscript.

    List of references  requires revision to comply with the journal's requirement.

Author Response

The authors manuscript on “Disorder” versus “Abuse”? Exploratory data on stigmatizing terminology among medical students at a Swiss University “  is a well written  with clearly defined  the topic and the results obtained in this  paper. However, it is the reviewer's responsibility to draw the authors' attention to aspects requiring revision / improvement:

Thank you for your positive comments and your suggestions on how to improve our manuscript. In response, we have now made the following changes:

Comment 1: First page: missing information about type of manuscript (review or article) above the title in left part of the page.

Response 1: We have now added that the manuscript is an “Article”, in line with your request.

Comment 2: It seems missing reference(s) numer(s) to support quoted definition of stigma in Box 1: “Definition of stigma”.

Response 2: We have added 3 references to support the definition. In addition, we have reviewed the citation of all references to ensure that they are in the correct numerical order and referencing format.

Comment 3: All abbreviation used first time in the main text must be explained. In example abbreviation “SUD” in line 73,74, 75 is explained in line 80 (“substance use disorder (SUD)”) but not in line 73.

Response 3: Thank you for drawing our attention to this. The suggested ammendment has been made and all abbreviations have been updated to ensure that the full title is provided before the acronym.

Comment 4: line 153-154, (quote) , “For our study we used the questionnaire developed by John F. Kelly and Cassandra 153 M. Westerhoff” must be supported by reference.

Response 4: A reference was included but it was initially placed in the description below this statement. For clarity, we have now removed it from its original position and placed the reference immediately after the statement, as per your suggestion.

Comment 5: Line 172-173, please be more specify about details of quoted ,, Initially, we contacted one of the authors to ask if they agreed that we could translate and use their questionnaire “. It is missing data : Who are the authors of questionnaire? What is name of questionnaire?

Response 5: We have now added a more specific explanation, including the reference for this work.

Comment 6: It is recommended to provide and attach questionnaire used in this study to manuscript.

Response 6: As per your suggestion, the questionnaire has now been added to Appendix 1 and the vignettes that were used have been added to Appendix 2. A comment to this effect has been added to the “Questionnaire” subsection (“Method” section).

Comment 7: List of references requires revision to comply with the journal's requirement.

Response 7: We have updated all references in MDPI format. We have also re-checked that all references appear in their correct numerical order.

Thank you, once again, for your feedback.

Reviewer 2 Report

This is a short exploratory study about the effects of "stigmatizing" terminology on medical students’ perceptions and attitudes toward substance use disorder (SUD) patients.

This is a timely and interesting topic, with relevant consequences for medical training and practice.

The methodological approach used is adequate and the size of the sample is large enough. Overall, results do not confirm the impact of "stigmatizing" terminology – although the authors argue (based on excessively fragile empirical support, I believe) for some trends...

Several methodological points should be considered, although they will probably not change the general conclusions of this study.

The goals of the study are not clear. "To explore the feasibility (...) of a future study on this topic” seems to be a goal necessarily achievable – in my opinion, assessing the feasibility of such a study does not merit a scientific paper. Acceptability is more interesting but I believe it is secondary to the main results, so it can be just addressed in the Discussion and not a goal in itself. So, I suggest the authors restrict themselves to the formulation of the second goal. The first goal is again repeated at the end of the Questionnaire subsection. Please, consider moving it to a more appropriate place.

The Method Section is somehow confusing. The Participants section presents information that I believe does not characterize the sample and is not a relevant methodological issue. Perhaps this information justifies the pedagogical and practical relevance of the study (namely, information about PROFILES and CanMEDS), so the authors should consider moving it to the Introduction or the Discussion.

The description of the questionnaire is too limited. Only "big" themes are listed ("cause of the problem", "social threat", "able to regulate his substance use" and "receive therapeutic vs. punitive actions"). Do all questions fall into these three themes? Are there more themes in the questionnaire? I believe it would be very useful for the reader to know the exact questions included in the questionnaire (for instance, presenting them in Table 2).

Why did the authors reduce six-level rating scale measures to dichotomous data (agree vs. disagree)? An analysis based on ranks (Mann-Whitney test) or means (t-test) could be more sensitive to the effect of the vignettes.

Data analysis is somehow weakened due to multiple testing: 32 chi-square tests increase steadily the risk of Type I errors (80% of the risk of at least one false-positive). Unfortunately, no significant results were obtained and in multiple testing situations, marginally significant results are almost certainly false positives and should be ignored. A possible alternative for a more formally correct solution: correcting for multiple testing (Benjamini and Hochberg False Discovery Rate, e.g.) or computing composite scores for themes (summing items scores) and testing differences on these thematic composites (and not separate by items).

An additional possibility is to consider university years (or, at least, contrasting Bachelor vs. Master levels) in the analysis, to explore possible trends along graduation training.

Concerning the acceptability of the study, I suggest the authors discuss/interpret the response rates and their increase through study years (from 13% to 45%)

When discussing response levels (for instance: “some participants considered the person with SUD to be responsible for their own health condition”, line 271), it would be interesting to know the corresponding percentages (I am not certain that this sentence in the Discussion corresponds to the result in line 235, so I just can speculate how many students expressed this attitude).

The differences the authors discussed (related to items 17, 23, 25, and 29) are small-sized (around 10% and the corresponding effects sizes, as measured by phi coefficient, are always small effects, according to guidelines defined by Cohen: Cohen J (1992) A power primer. Psychological Bulletin 112: 155–159). So, I am afraid that when authors state that “we were nevertheless able to explore some trends in the perceptions of future healthcare professionals” (line 282) or next paragraph (lines 290-296), they should be more cautious. The conclusion that “Nevertheless, we can hypothesize that these trends suggest the direct impact of language on perception, judgment, and therapeutic approach” (lines 302-303) or in lines 322-323 is not empirically well supported – more cautious can be used when arguing based on this potentially null result.

I believe even these null results are interesting – the indifference to stigmatizing terminology in a medical student sample is an interesting result. However, I ask the authors to consider comparing vignettes based on ranks or directly on ratings, to check if reliable differences emerge. I also suggest the authors explore the effect of the study year (or Bachelor vs. Master). Perhaps a small non-significant effect on the total sample would emerge on a specific subgroup (perhaps older students are more sensitive to terminology).

Minor: definitions of "stigma" (lines 37-44 and Box 1) need bibliographic support

The acronym SUD appears (line 73) before its definition (line 80)

The way the two versions of the questionnaire were randomly allocated to participants is not clear (lines 188-190).

It is not clear how a six-point rating scale has a level labeled “indifferent” (line 205). Please, clarify.

Does the described sociodemographic information (lines 223-224) correspond to the universe from where participants were recruited? If yes, I believe such information is not a result, so it should be provided in the Participants section.

Author Response

This is a short exploratory study about the effects of "stigmatizing" terminology on medical students’ perceptions and attitudes toward substance use disorder (SUD) patients.

This is a timely and interesting topic, with relevant consequences for medical training and practice.

The methodological approach used is adequate and the size of the sample is large enough. Overall, results do not confirm the impact of "stigmatizing" terminology – although the authors argue (based on excessively fragile empirical support, I believe) for some trends...

Thank you for taking the time to read our manuscript and give your detailed feedback. We fully agree with your perspective, and have adapted the manuscript so that it better reflects the issues that you have raised. Please find responses to each of your comments below:

Several methodological points should be considered, although they will probably not change the general conclusions of this study.

Comment 1: The goals of the study are not clear. "To explore the feasibility (...) of a future study on this topic” seems to be a goal necessarily achievable – in my opinion, assessing the feasibility of such a study does not merit a scientific paper. Acceptability is more interesting but I believe it is secondary to the main results, so it can be just addressed in the Discussion and not a goal in itself. So, I suggest the authors restrict themselves to the formulation of the second goal. The first goal is again repeated at the end of the Questionnaire subsection. Please, consider moving it to a more appropriate place.

Response 1: We fully agree with your comments and in response we have reformulated the aims and chosen to remove the repetition.

Comment 2: The Method Section is somehow confusing. The Participants section presents information that I believe does not characterize the sample and is not a relevant methodological issue. Perhaps this information justifies the pedagogical and practical relevance of the study (namely, information about PROFILES and CanMEDS), so the authors should consider moving it to the Introduction or the Discussion.

Response 2: Thank you for this feedback. We have considered this issue but we feel that the information on PROFILES and CanMEDS should remain in the participants section, to best explain our choice of population.

Comment 3: The description of the questionnaire is too limited. Only "big" themes are listed ("cause of the problem", "social threat", "able to regulate his substance use" and "receive therapeutic vs. punitive actions"). Do all questions fall into these three themes? Are there more themes in the questionnaire? I believe it would be very useful for the reader to know the exact questions included in the questionnaire (for instance, presenting them in Table 2).

Response 3: To avoid an information overload in Table 2., A copy of the questionnaire and the vignettes have now been added to Appendix 1 and 2. A comment to this effect has been added to the “Questionnaire” subsection (“Method” section).

With regard to the themes of the questions that we describe, we do believe this is a limitation of our study. In the original version Kelly and Westerhoff`s questionnaire, the authors had grouped the items in several other dimensions. Unfortunately, whist the authors gave their consent that we could translate and use their questionnaire, they never provided information on the distribution of items by dimensions or the key to analysis of these dimensions (although we did request this information from them several times). We therefore had to limit ourselves to a simpler, descriptive analysis of the data collected by our French version of the questionnaire, without being able to do the complete analysis of these data as in the original version. Being unable to conduct any further analysis, we had therefore acknowledged this issue in the “Strengths and limitations” subsection (“Discussion” section).

Comment 4-5: Why did the authors reduce six-level rating scale measures to dichotomous data (agree vs. disagree)? An analysis based on ranks (Mann-Whitney test) or means (t-test) could be more sensitive to the effect of the vignettes.

Data analysis is somehow weakened due to multiple testing: 32 chi-square tests increase steadily the risk of Type I errors (80% of the risk of at least one false-positive). Unfortunately, no significant results were obtained and in multiple testing situations, marginally significant results are almost certainly false positives and should be ignored. A possible alternative for a more formally correct solution: correcting for multiple testing (Benjamini and Hochberg False Discovery Rate, e.g.). or computing composite scores for themes (summing items scores) and testing differences on these thematic composites (and not separate by items)

Response 4-5: As explained in comment 3, regretfully the authors did not provide us with the analytical key used in their original article. We therefore had to limit ourselves to a descriptive analysis without analyzing the results by subscales as per Kelly and Westerhoff's article, and which was our original objective. However, in response to your feedback, we re-analysed these figures using the Mann-Whitney test and corrected some results.

Comment 6: An additional possibility is to consider university years (or, at least, contrasting Bachelor vs. Master levels) in the analysis, to explore possible trends along graduation training.

Response 6: We have now analyzed by comparing the courses (Bachelor`s or Master`s) and even compare each year of study between the two courses. As the groups are relatively small, we did not find any statistically significant differences between them. We have therefore added a comment to this effect within the “Results” section.

Comment 7: Concerning the acceptability of the study, I suggest the authors discuss/interpret the response rates and their increase through study years (from 13% to 45%)

Response 7: In response to your comment we have added a few sentences to describe the response rates and participant groups in more detail, within the “Results” section.

We have also added text to the “Strengths and limitations” paragraph (“Discussion” section), where we discuss/interpret the implications of such a varied response rate, within the study.

Comment 8: When discussing response levels (for instance: “some participants considered the person with SUD to be responsible for their own health condition”, line 271), it would be interesting to know the corresponding percentages (I am not certain that this sentence in the Discussion corresponds to the result in line 235, so I just can speculate how many students expressed this attitude).

Response 8: We added examples of the percentages to support the statements that you mentioned in your comment.

Comment 9: The differences the authors discussed (related to items 17, 23, 25, and 29) are small-sized (around 10% and the corresponding effects sizes, as measured by phi coefficient, are always small effects, according to guidelines defined by Cohen: Cohen J (1992) A power primer. Psychological Bulletin 112: 155–159). So, I am afraid that when authors state that “we were nevertheless able to explore some trends in the perceptions of future healthcare professionals” (line 282) or next paragraph (lines 290-296), they should be more cautious. The conclusion that “Nevertheless, we can hypothesize that these trends suggest the direct impact of language on perception, judgment, and therapeutic approach” (lines 302-303) or in lines 322-323 is not empirically well supported – more cautious can be used when arguing based on this potentially null result

Response 9: We fully agree with your comments about the difficulties in interpreting small-sized and non-significant effects from small samples. In response to this issue we have adapted several comments in the “Discussion” section, to consider what can be taken from our study, with greater caution.

Comment 10: I believe even these null results are interesting – the indifference to stigmatizing terminology in a medical student sample is an interesting result. However, I ask the authors to consider comparing vignettes based on ranks or directly on ratings, to check if reliable differences emerge. I also suggest the authors explore the effect of the study year (or Bachelor vs. Master). Perhaps a small non-significant effect on the total sample would emerge on a specific subgroup (perhaps older students are more sensitive to terminology).

 Response 10: As per our explanation in Response 6, We analyzed by comparing the courses (Bachelor`s or Master`s) and even compare each year of study between the two courses. As the groups are relatively small, we did not find any statistically significant differences between them.

Comment 11: Minor: definitions of "stigma" (lines 37-44 and Box 1) need bibliographic support

Response 11: Three bibliographic references have now been added to support the definition.

Comment 12: The acronym SUD appears (line 73) before its definition (line 80)

Response 12: All abbreviations have been checked and any necessary changes have been made to ensure that a full title appears before the first acronym.

Comment 13: The way the two versions of the questionnaire were randomly allocated to participants is not clear (lines 188-190).

Response 13: When a participant clicked on the link to answer the questionnaire, the limesurvey plateform randomly assigned one of the two versions of the questionnaire to them. We have therefore added a paragraph to this effect, which we hope clarifies the matter.

Comment 14: It is not clear how a six-point rating scale has a level labeled “indifferent” (line 205). Please, clarify.

Response 14: We added an explanation to the “Statistical analysis” subsection (“Method” section) to describe the use of the “indifferent” category.

Comment 15: Does the described sociodemographic information (lines 223-224) correspond to the universe from where participants were recruited? If yes, I believe such information is not a result, so it should be provided in the Participants section.

Response 15: This information was provided by the University, so as per your suggestion we have removed the relevant sentence from the “Results” section and placed it in the “Participants” subsection (“Method” section).

We hope that we have addressed the issues that you have raised, to your satisfaction and we thank you, once again for having taken the time to give your feedback on our manuscript.  

Round 2

Reviewer 2 Report

I thank the authors for their effort in adjusting their manuscript to my previous comments. Overall, I'm satisfied with the answers given and the present version of the text. However, I still list a series of suggestions that being minimal, I believe will improve the readability of the manuscript. Importantly, although the lack of terminology effect is the most evident finding of this study, the authors do not provide a straightforward interpretation for this insensitivity to terminology in medical students. I believe this would be an important piece in the Discussion.

Line 54 – Please, correct the omission of the white space in "hasbeen redefined".

lines 55-59 - Being a citation, I believe it is necessary to indicate the page where it is taken from.

Lines 60-69 – I believe a bibliographic reference is needed here.

Line 70 - Please, consider replacing "it" with "stigma", to make the sentence easier for the reader.

Lines 135-151 - I still think that presenting the new learning context of medical students (as the way it is presented in the manuscript) does make much sense in the Participants Section, especially when the last paragraph corresponds to a new paraphrase of the aim of the study ("to evaluate the current students' interest in the subject and their perceptions, to raise awareness of the implications of terminology in their future practice"). Perhaps this is a rather formal topic and I leave the authors the option to do what they think is more adequate for their proposes. However, I believe these ideas are important for the justification of the study (so, they should be integrated into the Introduction’s last paragraph). Alternatively, if the authors persist on leaving this information in the Participants’ Section, please consider rephrasing in the format of an explicit justification for using this specific population for the study.

Line 187 - Please, correct the typo" palteform". Please, use the upper letter case in "limesurvey". Also, please check the letter font for that sentence. Finally, if the actor of the sentence is "a participant", I believe the personal pronoun should not be "them " but instead "him or her".

Line 208 - It is not clear how the Mann-Whitney test was used if the authors kept the data in the dichotomic format (agree vs. disagree).  My previous suggestion was to analyze the original rating scores (from 1 to 6) or even compositive scores (aggregating questions). In such a case, using the Mann-Whitney test will be an adequate analytic approach. However, if the authors kept the dichotomic data format, I believe they should return to their Chi-square or the Fisher exact tests, as presented in the previous version of the manuscript. So, please, consider reporting the original p-values (instead of the p-values taken from Mann-Whitney tests since, in my opinion, make no sense). Sorry for the confusion.

Line 213 – Please, verify if the empty parentheses are necessary.

Table 1 - Please, check the alignment of the first percentage on the table (28.35)

Lines 226-228 - I did not understand the sentence (namely, the second part), Please, clarify. Please, check the letter font for this sentence.

Lines 258-260 – The authors said: “Several further analyses were conducted to compare the Bachelor`s and Master`s groups, and even compare each year of study between the two courses.” First, what is meant by “courses”? Master vs. Bachelor? I believe such terminology was not used before in the manuscript. Second, which type of analysis was run? If the authors kept the dichotomic format for data, I suggest that a split analysis should be run: the same chi-square association test for vignette (A vs. B) and answers (agree vs. disagree) but run separately for Bachelor’s and Master’s students. I hoped that older students were more sensitive to wording and that a clear association would appear among Master’s students. As it is, the sentence does not clarify the analytic procedure followed.

Lines 271-281 – Please, check the letter font size.

Line 276-278 and ff - Please, consider indicating the number of the question. For instance: "For example, when they were asked if Mr. Williams' problem was caused by his bad choices (question 3), 44% (vignette A) versus 47% (vignette B) of participants agreed." This information will guide the reader if he wants to check the figures in Table 2 or the integral text of the question in Appendix 2. The same suggestion is for lines 280-281, lines 301-303, line 306, and line 309. I believe it is important for the reader to know the exact question that the authors are discussing.

Line 292 – What do the authors intend to say with “The original study we used…”? I do not understand the meaning of “using a study”. Please, rephrase.

Line 298 – What do the authors mean by “did not observe clinically significant differences”?

Line 301-303 – I felt a contradiction between the last sentence (emphasizing the “trends”) and the previous ones (minimizing the trends: “the 2 groups tend to have the same opinion” or “there was almost unanimous agreement between the participants of both groups” or “the trends were not pronounced”). To mitigate such contradiction, I suggest the final sentence be toned down with something like: "Although not being statistically reliable in our sample, we can hypothesize that these trends...".

Appendix 2 – In the appendix, vignettes are called “Vignette 1” and “Vignette 2”, while in the main text, they appeared as “Vignette A” and “Vignette B”. Perhaps using the same label would be easier for the reader.
